# Metformin: Sex/Gender Differences in Its Uses and Effects—Narrative Review

**DOI:** 10.3390/medicina58030430

**Published:** 2022-03-16

**Authors:** Ioannis Ilias, Manfredi Rizzo, Lina Zabuliene

**Affiliations:** 1Department of Endocrinology, Diabetes and Metabolism, Elena Venizelou Hospital, GR-11521 Athens, Greece; 2Department of Health Promotion Sciences, Maternal and Infantile Care, Internal Medicine and Medical Specialties (Promise), School of Medicine, University of Palermo, Via del Vespro, 141, 90127 Palermo, Italy; manfredi.rizzo@unipa.it; 3Faculty of Medicine, Vilnius University, M. K. Čiurlionio St. 21/27, LT-03101 Vilnius, Lithuania; lina.zabuliene@mf.vu.lt

**Keywords:** Metformin, gender, insulin resistance

## Abstract

Metformin (MTF) occupies a major and fundamental position in the therapeutic management of type 2 diabetes mellitus (T2DM). Gender differences in some effects and actions of MTF have been reported. Women are usually prescribed lower MTF doses compared to men and report more gastrointestinal side effects. The incidence of cardiovascular events in women on MTF has been found to be lower to that of men on MTF. Despite some promising results with MTF regarding pregnancy rates in women with PCOS, the management of gestational diabetes, cancer prevention or adjunctive cancer treatment and COVID-19, most robust meta-analyses have yet to confirm such beneficial effects.

## 1. Introduction: Metformin—Gender Medicine

Metformin (MTF) occupies a major and fundamental position in the therapeutic management of type 2 diabetes mellitus (T2DM) [1,2,3]. Sex pertains to “the different biological and physiological characteristics of males and females, such as reproductive organs, chromosomes or hormones”, whereas gender pertains to “the socially constructed characteristics of women and men—such as norms, roles and relationships of and between groups of women and men", quoting the relevant definitions from the Council of Europe (https://www.coe.int/en/web/gender-matters/sex-and-gender, accessed on 7 March 2022). Gender medicine is the medical discipline that integrates any effect of sex and/or gender on the overall level of health (prevention, diagnosis and treatment/management of diseases), taking into account biological as well as social sex differences [4]. Its aim is to improve health for any gender. Gender medicine may be a neglected dimension of medicine. Research results are accumulating, pointing to sex/gender-related differences in prescribing, as well as in the pharmacokinetics, the pharmacodynamics, the efficacy and side effects of various medications. In this concise review we will attempt to present the impact of sex/gender on the therapeutics of MTF. For practical purposes, in the following text, we will have to refer to human studies with the terms “sex” and “gender” interchangeably, acknowledging that this may not be correct; we will refer to animal studies with “sex” only.

## 2. Pharmacokinetics, Pharmacodynamics and Metabolism of MTF

MTF is a weak base, and is very polar and extremely soluble in water [5]. It is absorbed from the small intestine, leading to a peak in concentration in one to two hours after oral intake. Its bioavailability is from 50% to 60% [6]. MTF is weakly bound to proteins. Its plasma half-life is estimated to be 1.5 to 5.0 h and it is practically unmetabolized after being distributed mainly in the liver, kidneys and intestine [7,8,9]. Excretion occurs via the kidneys, with a clearance of 933–1317 mL/min, involving glomerular filtration and tubular secretion [9]. The mechanisms of cellular action of MTF are still poorly understood. Various molecular responses are elicited by MTF and apparently some are influenced by sex hormones (Figure 1) [10,11,12,13]. The normoglycemic effect of MTF results mainly from a decrease in hepatic glucose production by inhibition of gluconeogenesis and by an action on glucose-6-phosphatase [2]. In addition to this action on the liver, which results mainly in a decrease in fasting blood sugar, MTF also potentiates the effect of insulin on muscle glucose uptake.

## 3. Gender-Specific Use of MTF

### 3.1. MTF in Women with Polycystic Ovary Syndrome

Polycystic ovary syndrome (PCOS) is a common endocrine disease (with variable prevalence worldwide, ranging from 6% to 26%) [14], which is characterized by anovulation, clinical or biochemical hyperandrogenemia and polycystic ovary morphology [15]. Hyperinsulinemia as a result of tissue insulin resistance, is central to PCOS [16]. Insulin resistance is observed in 45–65% of patients with PCOS and is associated with excessive phosphorylation of insulin receptors. Hyperinsulinemia, impaired glucose tolerance, dyslipidemia and hypertension affect 40–45% of patients with PCOS [17]. Hyperinsulinemia adversely affects the hypothalamic-pituitary-ovarian axis, resulting in altered endocrine control, menstrual irregularity and infertility [15].

Many interventional studies have demonstrated the positive effect of MTF on both the reproductive as well as metabolic aspects of the syndrome [15]. However, the mechanisms, by which MTF exerts its effects in treating PCOS, are only partially understood. The rationale of MTF use in PCOS is based on the fact that hyperinsulinemia is the basis of the syndrome and adversely affects ovarian function. Insulin boosts 17OH-progesterone activity causing ovarian stroma hypertrophy, follicular atresia and anovulation [15]. Therefore, MTF directly or indirectly improves steroidogenesis (this has been noted by in vitro studies of granuloma cell response to follicle stimulating hormone (FSH) and insulin growth factor 1 (IGF-1) [18]).

In older studies, MTF use was associated with normalization of all insulin resistance parameters, in all PCOS patients grouped according to body mass index (BMI), and with degree of insulin resistance [19,20]. It was also shown that MTF led to improvement in BMI, diastolic blood pressure and high-density lipoprotein (HDL) cholesterol levels, decreasing the prevalence of metabolic syndrome in women with PCOS by 34.3% to 21.4%, in a dose-dependent fashion [21]. The usual MTF dosage described in the literature starts from 500 mg/day and reaches up to 850 mg three times daily for a duration of at least six weeks [22,23]. In 2013, the American Endocrine Society issued guidelines on the management of PCOS which included MTF as a treatment [24]. Specifically, MTF was recommended for women with PCOS and T2DM or insulin resistance, after failure of lifestyle change, diet and exercise, as a daily routine. It was not advised as a first line treatment of skin manifestations of PCOS (hair loss, acne), complications of the syndrome in pregnancy or for obesity [15]. MTF can also be given to women with menstrual disorders in which contraceptive treatment has failed or in women who wish to have children, as a second choice of treatment. There is no straightforward answer to whether all women with PCOS should undergo MTF treatment [25]. Proponents of MTF consider it a necessary drug for women with PCOS not only to prevent its long-term complications (in the context of insulin resistance) but also because MTF has been shown to improve all of the syndrome’s parameters. The first line treatment that includes diet and physical exercise is a time-consuming process that requires the compliance of women on a strict, long-term schedule, and relapse is very common.

Another element that comes to add to the beneficial action of MTF is in the treatment of adolescent girls with obesity and hyperandrogenemia. It seems that 50% of adolescent women with hyperandrogenemia have already developed resistance to progesterone-mediated gonadotropin-releasing hormone (GnRH) pulse suppression. The abnormal regulation of GnRH and luteinizing hormone (LH) secretion by the persistence of increased frequency of GnRH pulses is already present in adolescent girls with hyperandrogenemia before menarche [26]. Thus, the correction of androgen overproduction in PCOS is deemed to be necessary. The administration of MTF for at least three months improves glucose tolerance, lowers testosterone levels and plasma insulin and reduces adrenal overproduction of androgens in obese adolescents [27]. On the other hand, there are many who dispute the efficacy and benefits of MTF as a permanent treatment for women with PCOS. Insulin resistance is common but is not the main feature of PCOS [28,29]. Insulin sensitivity varies with the phenotype of the syndrome. MTF seems to improve the effects of hyperandrogenemia, though to a lesser degree than that of antiandrogens or oral contraceptive pills (OCP); therefore, MTF—according to some experts—cannot be used as a first-line treatment for these cases. Some also argue that insulin resistance cannot be reliably diagnosed from surrogate indicators. Insulin levels reflect its secretion as well as its clearance and do not adequately predict its action [30]. 

In the most recent and extensive relevant meta-analysis, treatment with MTF only was found to improve rates of live births when it was compared to no treatment (odds ratio [OR] = 1.59, 95% confidence interval [CI] = 1.00-2.51; 4 studies with total *n* = 435; the authors of the meta-analysis, however, found the quality of the data to be of low quality) [31]. Twenty-two percent to 40% of women experienced gastrointestinal side effects. Rates of ovulation and clinical pregnancy with MTF alone were higher compared to those with placebo (with OR of 2.64 [95% CI = 1.85-3.75] and 1.98 [95% CI = 1.47-2.65], respectively) [31]. In the same meta-analysis, no firm conclusions were formulated regarding MTF versus clomiphene citrate treatment, although the combination of both medications may increase ovulation rates and rates of clinical pregnancy [31]. The reproductive effects of MTF may be lower in obese compared to non-obese women. The same meta-analysis did not find an effect of MTF, when used alone, on the BMI of women with PCOS when compared to placebo; a probable reduction in BMI was noted for the combination of MTF with clomiphene citrate [31]. Furthermore, no firm conclusions were reached regarding MTF and testosterone, glucose or insulin levels in women with PCOS.

### 3.2. MTF in Women with Gestational Diabetes

When used in pregnancy, MTF does not show appreciable changes in its bioavailability, because any changes in the latter are being offset by changes in MTF’s clearance [32]. Although the mainstay of drug therapy for gestational diabetes (GDM) is insulin, MTF is used in pregnancy with increasing frequency, though it is still limited to a maximum of 5–6% of all medications for GDM, depending on the country in which it is assessed [33]. The use of MTF in pregnancy is considered to be safe overall, with favorable effects on maternal weight gain, the incidence of preeclampsia, the dosage in concomitant insulin administration, and the rate of fetal macrosomia and neonatal hypoglycemia; it may increase the rate of small-for-gestational-age infants [34,35,36,37,38,39,40]. Apparently, MTF lowers proinflammatory cytokines (tumor necrosis factor alpha (TNF-α), interleukin [IL]-1-alpha and IL-1-beta and IL-6) in serum, placenta and omental tissues [41]. Interestingly, although in a very critical assessment of metanalyses regarding MTF, most were deemed to be of low quality, the exception being those in obstetric/gynecological settings [42]. There are caveats in the use of MTF in pregnancy: it was found—in vitro, in human embryonic stem cells—to decrease the differentiation of pancreatic beta cells [43], and in mice to decrease or arrest early embryonic development [44].

## 4. Sex/Gender Differences Using MTF

### 4.1. Prescribing/Administering MTF for Diabetes

There may be a difference in T2DM prevalence by sex/gender [45]; this difference depends on the definition of diabetes per se: men tend to have higher fasting plasma glucose more often, whereas women tend to have abnormalities in the oral glucose tolerance test (both modalities are used in the diagnosis of the disease) [46]. Although MTF is widely prescribed worldwide as a first-choice medication for T2DM [47], few studies have seen the light regarding use by gender. Although women are more concerned than men about their body image [48], and MTF may show a modest effect on weight loss [49], women are usually prescribed lower MTF doses compared to men (and they report more gastrointestinal side effects) [50]. In a Dutch study, although women reported/experienced more gastrointestinal drug reactions during the first months of MTF treatment, the rate of the latter dropped in a way analogous to that in men [50]. Furthermore, after nine months of treatment, women were given a significantly lower daily dose. The caveat is that the researchers did not correct for body weight or body mass index (which is different between men and women) [51]. In one study, women in Austria were more apt to be prescribed MTF compared to men [52], whereas in another study, women in New Zealand were more apt to discontinue this treatment due to mainly gastrointestinal side effects [53]. In a recent report, with data from the Metformin and AcaRbose in Chinese as the initial Hypoglycemic treatment (MARCH) study, fasting and 2-h postprandial glucose were lower in women on MTF compared to men on MTF after 24 and 48 weeks of treatment. In the same study, an increase in insulin secretion was noted in women treated with MTF, whereas no appreciable change was noted in men on MTF [54]. Interestingly, in a fairly recent study, which was hampered by the small number of participating women in it (n = 13), the participants could not distinguish between MTF and placebo and did not report more gastrointestinal side effects compared to placebo [55]. While serum creatinine can be used as a criterion for use or non-use of MTF, a few years ago the focus shifted towards the glomerular filtration rate [measured (GFR) or calculated (eGFR)] as a criterion [56,57]. With such a criterion, more patients can be given MTF compared to those that can be given it using serum creatinine as a criterion, provided that eGFR is over 45 mL/min/1.73 m^2^. Additionally, patients already on MTF with eGFR in the 30–44 mL/min/1.73 m^2^ range can continue treatment at a maximum dose of 1 g/day. This change in criteria can have a profound effect on increasing the size of the target group of patients to be administered MTF, since gender differences in GFR/eGFR can lead to variance in prescription patterns [56]. Women have lower GFR compared to men and show a higher decline in this parameter of kidney function with advancing age. Thus, in this light, fewer women—particularly older ones—may be given MTF compared to men [56,58].

### 4.2. MTF and Vitamin B12/Homocysteine

The long-term administration of MTF significantly lowers vitamin B12 levels [59,60,61]. Vitamin B12 deficiency with MTF is rarely symptomatic; it is linked to a reduction in the intestinal absorption of cobalamin and can be reversed by the discontinuation of MTF or with oral B12 supplementation. Men have lower vitamin B12 levels compared to women [62]. In a study of patients with T2DM (without a control group), higher doses of MTF and male sex were factors associated with lower levels of vitamin B12 [63]. Nevertheless, the effect of MTF on B12 by sex/gender, to the best of our knowledge, has not been assessed adequately; this is of interest given the sex/gender differences presented above. Additionally, MTF may conditionally elevate or reduce homocysteine levels, which is critical for people with obesity [64,65].

### 4.3. MTF and Cardiovascular Disease

MTF is considered to be associated with some degree of cardioprotection [66,67]; the latter is apparently the net result of its beneficial actions on endothelial and smooth muscle cells, blood lipids and systemic chronic inflammation [68,69]. In experimental models, MTF was beneficial with regards to myocardial reperfusion, fibrosis and inflammation in post-experimental myocardial ischemia [70,71]. In an older, small scale, study, MTF was noted to have a favorable effect on cardiac metabolism in women (increasing myocardial glucose uptake and lowering fat metabolism), in contrast to having an unfavorable one (with opposite effects) in men [72]. In a study of 167,254 (46% women) patients with T2DM who were already using MTF and started newer anti-diabetic medications, the incidence of cardiovascular events, after a median observation time of 4.5 years, in women was lower compared to that of men (14.7 versus 16.7 per 1000-person-year) [73]. Nevertheless, a systematic review of MTF’s overall actions has not been conclusive regarding micro- and macrovascular complications in patients with T2DM [74].

### 4.4. MTF and Andrology/Urology

In small (and—apparently—underpowered) studies, the effects of MTF solely on men have been observed. This medication has been reported to be of benefit in non-diabetic men with erectile dysfunction who had not responded to sildenafil [75]. The mechanisms are obscure: they may be direct, via endothelium-dependent vasodilatation or attenuation of sympathetic nerve activity, and indirect, via MTF’s effect on blood pressure [75]. Indirect proof of the low power of studies is that erectile dysfunction, low sex drive and low testosterone (total, free and bioavailable) have also been attributed to MTF use in men with T2DM [76]. Furthermore, the use of MTF for T2DM, in men with prostate cancer, has been associated with lower prostate-specific antigen levels and improved survival [75,77,78,79].

### 4.5. Musculoskeletal Effects of MTF

From in vitro studies, a role for MTF has been proposed in the stimulation of osteogenesis; in vivo studies are less conclusive [80]. Furthermore, MTF activates adenosine monophosphate-activated protein kinase (AMPK) signaling pathways. The activation of AMPK has been implicated in muscle repair [80]. Thus, it is not surprising that patients using MTF report less musculoskeletal pain vis-à-vis patients not on MTF [81]. The beneficial musculoskeletal effects of MTF were recently found to be more pronounced in women compared to men [81].

### 4.6. MTF and Experimentally-Induced Neurological Disease

An interesting dimorphism has been observed in mice regarding experimentally induced neuropathic pain and spinal cord microglial activation. MTF was shown to prevent and reverse neuropathic pain and spinal cord microglial activation only in male mice [82]. The researchers presume that the known activation of AMPK may be implicated, although no firm etiology for the sex difference in observations has been formulated. On the other hand, in another experimental study of brain injury in mice, MTF was beneficial for cognitive recovery in females but not males [83], pointing to a crucial relevant role for estradiol/testosterone [83].

### 4.7. MTF and Aging/Life Span (Experimental)

Dimorphic sex responses to MTF regarding life span have been described: chronic administration of MTF extended the lifespan of female mice and curtailed the lifespan of male mice [84]. Yet, more recent studies show that the positive effect of MTF on longevity is more prominent in male mice [85]. In the Mexican fruit fly the effect of MTF on longevity is dose-dependent, and is beneficial in higher doses for females and in lower doses for males [86].

### 4.8. MTF and Cancer

In older and newer studies, MTF in subjects with T2DM (in the older studies at low doses of 500 mg/day or less) was shown to be more beneficial vis-à-vis the incidence of colorectal cancer in women compared to men [87,88]. In men, MTF use may lower the risk of prostate cancer, but the effect—if any—is apparently slight and statistically non-significant [89]. In a Lithuanian cohort, the lowest risk for endometrial cancer was observed in diabetic women who used only MTF (with a standardized incidence ratio [SIR] of 1.69 and 95% confidence interval [CI] of 1.49 to 1.92) [90]. MTF was found to lower the markers of proliferation in endometrial cancer cells [91,92]. Regarding the treatment of endometrial hyperplasia (considered to be a precancerous entity) or of endometrial cancer with MTF, either alone or in combination with megestgrol/medroxyprogesterone acetate, there were some promising results stemming from small studies [93,94]. In an older meta-analysis, a beneficial effect of MTF on overall mortality in women with endometrial cancer was noted [95]. Nevertheless, in another meta-analysis, no beneficial effect of MTF was found regarding the progression of endometrial hyperplasia to cancer, histology, or rates of hysterectomy [96]. MTF is considered to lower the risk for breast cancer in subjects with diabetes [97]. Results of trials aiming at the prevention of breast cancer with MTF are pending [98]. A higher cumulative MTF dose decreases kidney cancer risk in T2DM patients [99]. An analysis of MTF by cumulative dose showed significantly lower mortality risk in the highest cumulative dose category (with hazard ratio [HR] of 0.76 and 95% CI of 0.58 to 0.99) [100].

### 4.9. MTF and the Microbiome

Subtle differences have been reported in the gut microbiome between the male and female offspring of MTF-treated mice [101], as well as after MTF treatment in adult mice [102]. Before treatment with MTF and a high-fat diet (HFD), female mice had a preponderance of *Lactobacillus* species, whereas male mice had a preponderance of *Proteobacteria* species. After ten weeks of HFD and MTF, the bacterial species were different in males and females: a more pronounced increase in *Bacteroides* was noted in female mice compared to male ones [102]. In this respect, Lee et al. [103] suggested that gut microbiota could be affected by hormone levels, subsequently influencing glucose and lipid metabolism [104]; one study demonstrated that progesterone promotes the growth of oral *Bacteroides* species [105]. Although various studies have demonstrated a positive relationship between abundance in *Bacteroides* species and the therapeutic effect of MTF, future studies should consider the influence of sex on the effect of hormones on *Bacteroides* [106]. Unfortunately, in a very recent human trial of MTF administration, which led to tangible changes in the participants’ gut microbiome, the authors give no details vis-à-vis sex/gender [107].

### 4.10. MTF and COVID-19

Currently, there is a global effort to fight and win against the new severe acute respiratory syndrome coronvirus-2 (SARS-CoV-2) pandemic and its related coronavirus disease 2019 (COVID-19); proper management of T2DM is of even greater importance, since the presence of diabetes is associated with the most severe forms of COVID-19 and related mortality [108,109], and glycemic control is crucial [110,111]. In addition, a significant increase of cardiometabolic complications has been reported in many geographical areas, highlighting the need of a comprehensive and multidisciplinary approach to this terrible pandemic [112,113]. Insights and lessons from this experience can guide us to better manage cardiometabolic risk and overcoming current challenges [114,115]. In this context, the action and the effects of distinct antidiabetic drugs, including MTF, have been extensively investigated over the last two years [116,117], as well as the impact of genetics, comorbidities and inflammation on gender differences in COVID-19 outcomes [118].

There are many studies investigating MTF and COVID-19, and in particular mortality from this disease; the studies point to a beneficial (lowering) effect on mortality, but most did not report results by sex/gender of MTF users [119,120,121,122]. A large-scale study of mortality attributed to COVID-19 vis-à-vis MTF therapy [123] used anonymized data of patients with T2DM and/or obesity from a healthcare provider in the USA. In this study, the researchers compared a cohort of 3923 patients with COVID-19 not on MTF (55% women) to 2333 patients with COVID-19 on MTF (48% women). From the subgroups analyses it was found that women treated with MTF had a lower OR regarding mortality of 0.79 (95% CI: 0.64-0.98). The authors acknowledge that in their data no information about adherence to treatment with MTF was available. Furthermore, they also acknowledge that men are at higher risk of dying from COVID-19 and that men treated with MTF did not show any advantage in survival. An analysis of 328 patients’ data from China showed that MTF use was associated with a lower incidence of acute respiratory distress syndrome (ARDS) in women, whereas such an association was not been observed in men [124]. The authors speculated that the observed beneficial effect of MTF on women may have been the combined result of female sex and that MTF provided protection against the production of proinflammatory cytokines such as IL-6, IL-10 or TNF-α, which are known to be produced in abundance in COVID-19 [124]. In a study of an ex vivo animal model, MTF was shown to enhance the integrity of the pulmonary endothelial barrier [125]. The proposed mechanism involves the activation of AMP-activated protein kinase α1 (AMPK-α1, which in turn induces the activation of myosin light chain 2 (MLC2) and the deactivation of cofilin (a binding protein that regulates actin filament dynamics and depolymerization), supporting endothelial integrity [125]. A research group from the United States conducted a retrospective electronic health record data analysis of 25,326 subjects and reached contradictory results [126]. They found that the OR of dying remained significantly lower in male subjects on MTF (OR 0.28; 95% CI 0.09–0.88; *p* = 0.029) [126]. Experts have put forth many theories to explain why mortality rates in men were more than two-fold higher than in women. These theoretical assumptions include the different plasma concentrations of sex steroids, the differences in adipose tissue distribution between men and women, differences in the levels of circulating pro-inflammatory cytokines, and the known differences in the innate and adaptive immune responses to viral infections between sexes [126,127,128].

## 5. Conclusions

There are sex/gender differences with regards to glucose metabolism and the appearance of diabetes [129,130]. Pharmacogenetic studies have provided explanations—in part—for the variability and effectiveness in lowering glycemia with MTF [131]. To the best of our knowledge, sex/gender-wise, no such differences in the relevant genetic background have been shown to date. This may be a domain for future studies. Sex/gender differences in some effects and actions of MTF have been reported. Women may be prescribed lower MTF doses compared to men and report more gastrointestinal side effects. The incidence of cardiovascular events in women on MTF has been found to be lower to that of men on MTF. Despite some promising results with MTF regarding pregnancy rates in women with PCOS, for the management of gestational diabetes, cancer prevention or adjunctive cancer treatment and COVID-19, data from the most robust meta-analyses of clinical studies have yet to confirm such beneficial effects [35,38,132,133,134,135,136,137,138,139,140,141,142,143,144,145,146]. A caveat is that any extrapolation of benefit to humans from animal or in vitro studies has to take into account the level of MTF concentration attained; the latter may differ considerably and be lower in humans [147]. Vitamin B12 deficiency with long-term administration of MTF is tangible and needs to be tackled. Inconsistencies in the studies have been noted and the field is open for new research before implementation in clinical practice. Any beneficial effect of MTF—other than on glycemia in patients with T2DM—should be scrutinized in the absence of T2DM.

## Figures and Tables

**Figure 1 medicina-58-00430-f001:**
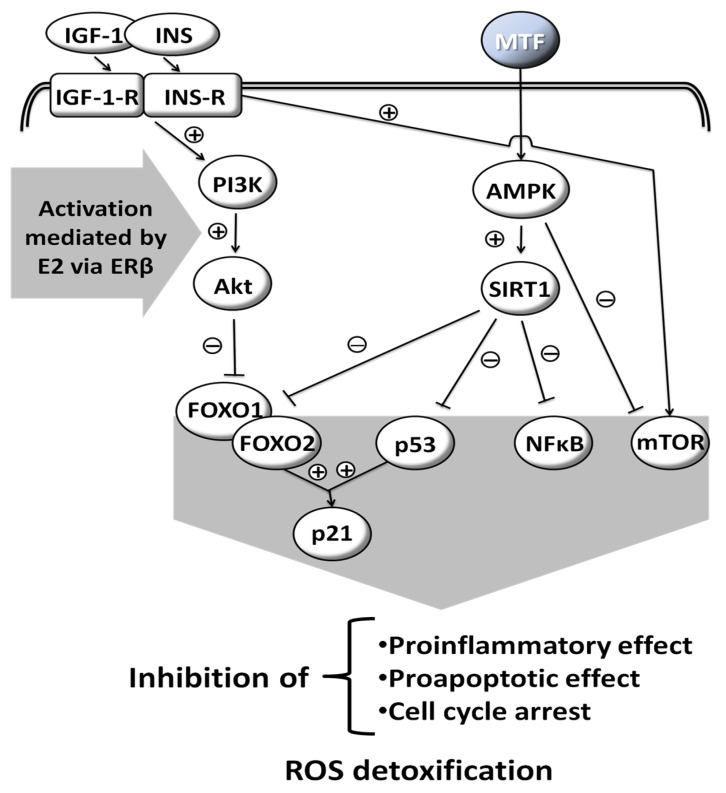
Molecular responses that are elicited by MTF; some are influenced by sex hormones. Insulin-like Growth Factor-1, INS: Insulin, IGF-1-R: Insulin-like Growth Factor-1-Receptor, INS-R: Insulin Receptor, PI3K: Phosphoinositide 3-kinase, Akt: Serine/threonine protein kinase B, AMPK: AMP-activated protein kinase, SIRT1: Sirtuin 1, FOXO1: Forkhead box protein O1, FOXO2: Forkhead box protein O2, p53: Tumor protein p53, NFκB: Nuclear factor kappa-light-chain-enhancer of activated B cells, mTOR: Mammalian target of rapamycin, p21: Cyclin-dependent kinase inhibitor 1, ROS: Reactive oxygen species, E2: Estradiol, ERβ: Estrogen receptor β, (+): activation, (−): inhibition, drawn with data from [10,11,12,13].

## Data Availability

Not Applicable.

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
