# Peer review of "Metformin: Sex/Gender Differences in Its Uses and Effects—Narrative Review"

_medicina, 2022, doi:10.3390/medicina58030430_

Round 1

Reviewer 1 Report

Introduction requires a thorough explanation of "gender medicine." Throughout the manuscript the term "gender" is used inappropriately: mice don't have genders (murine studies cited) and hormones aren't determined by gender. If "gender" is the focus of review, then all murine studies should be removed. Line 208 should read "sex drive" as it appears they are referring to libido.

Throughout, paper requires English language editing.

Much of review is just a summary of literature with little/no insight from authors. However, in section 4.2 the authors offers an interesting idea, but requires more discussion.

Line 300-304 authors indicate they believe reduced incidence of ARDS is result of female gender. Why and how? No explanation beyond listing a couple cytokines (which are present in males).

Conclusion should requires significant revision. More than just "inconsitencies in studies have been noted" is required to explain how and why "gender" plays a significant role in the use of MTF. Future directions? Studies that are required?

Author Response

We thank the Reviewers and Editors for the time and effort spent on assessing our work and for the insightful comments.

Responses to Comments from Reviewer  #1

[1]. Introduction requires a thorough explanation of "gender medicine." Throughout the manuscript the term "gender" is used inappropriately: mice don't have genders (murine studies cited) and hormones aren't determined by gender. If "gender" is the focus of review, then all murine studies should be removed. 
In the introduction of the revised version of the manuscript we have added that sex refers to  the different biological and physiological characteristics of males and females, such as reproductive organs, chromosomes, hormones, etc., while gender refers to the socially constructed characteristics of women and men – such as norms, roles and relationships of and between groups of women and men. 

[2]. Line 208 should read "sex drive" as it appears they are referring to libido.
The Reviewer is absolutely right. This has been corrected in the revised version of the manuscript.

[3]. Throughout, paper requires English language editing.
The paper’s grammar, spelling, punctuation and phrasing were reviewed.

[4]. Much of review is just a summary of literature with little/no insight from authors. However, in section 4.2 the authors offer an interesting idea, but requires more discussion.
Whereas a review of literature is a preparatory activity, a review paper is a culmination of knowledge and understanding on a given topic. Unfortunately, the data on metformin, beginning from its mode of action, are still unclear. We have revised slightly/appropriately section 4.2.

[5]. Line 300-304 authors indicate they believe reduced incidence of ARDS is result of female gender. Why and how? No explanation beyond listing a couple cytokines (which are present in males).
The section on ARDS and metformin has been enriched and revised; also a new recent reference has been cited. 

[6]. Conclusion [should] require[s] significant revision. More than just "inconsi[s]tencies in studies have been noted" is required to explain how and why "gender" plays a significant role in the use of MTF. Future directions? Studies that are required? 
The Conclusion has been reshaped as follows in the revised version of the manuscript: “There are sex/gender differences with regards to glucose metabolism and appearance of diabetes [129,130]. Pharmacogenetic studies, have provided explanations – in part – for the variability and effectiveness in lowering glycemia with MTF [131]. To the best of our knowledge, sex/gender-wise, no such differences in the relevant genetic background have been shown to date. This may be a domain for future studies. Sex/gender differences in some effects and actions of MTF have been reported. Women may be prescribed lower MTF doses compared to men and report more gastrointestinal side effects. The incidence of cardiovascular events in women on MTF has been found to be lower to that of men on MTF. Despite some promising results with MTF regarding pregnancy rates in women with PCOS, for the management of gestational diabetes, cancer prevention or adjunctive cancer treatment and COVID-19, data from the most robust meta-analyses of clinical studies have yet to confirm such beneficial effects [35,38,132-146]. A caveat is that any extrapolation of benefit to humans from animal or in vitro studies has to take into account the level of MTF concentration attained; the latter may differ considerably and be lower in humans [147]. Vitamin B12 deficiency with long-term administration of MTF is tangible and needs to be tackled. Inconsistencies in the studies have been noted and the field is open for new research before implementation in clinical practice. Any beneficial effect of MTF – other than on glycemia in patients with T2DM – should be scrutinized in the absence of T2DM.”

Reviewer 2 Report

Overall, this is a reasonably well-written manuscript reviewing sex differences in metformin treatment. I have a couple of minor comments:

  1. Line 55 - please add the prevalence estimates for polycystic ovary syndrome globally or at least in Europe.
  2. Line 152 – women are prescribed lower MTF doses than men – is this true even after correction for body weight?

Author Response

We thank the Reviewers and Editors for the time and effort spent on assessing our work and for the insightful comments.

Responses to Comments from Reviewer  #2
[1]. Line 55 - please add the prevalence estimates for polycystic ovary syndrome globally or at least in Europe.
This information has been added in the revised version of the manuscript.

[2]. Line 152 – women are prescribed lower MTF doses than men – is this true even after correction for body weight?
This is a very pertinent remark and we thank the Reviewer for it. We have added the following in the revised version of the manuscript: “The caveat is that the researchers did not correct for body weight or body mass index (which is different between men and women)”

Round 2

Reviewer 2 Report

Comments have been satisfactorily addressed.